# A Qualitative Consolidated Framework for Implementation Research Evaluation of Innovative PrEP Delivery During COVID-19 Among Adolescent Girls and Young Women in North West Province, South Africa

**DOI:** 10.3390/ijerph22101602

**Published:** 2025-10-21

**Authors:** Lerato Lucia Olifant, Edith Phalane, Yegnanew A. Shiferaw, Hlengiwe Mhlophe, Refilwe Nancy Phaswana-Mafuya

**Affiliations:** 1South African Medical Research Council/University of Johannesburg Pan African Centre for Epidemics Research Extramural Unit (SMRC/UJ PACER), Faculty of Health Sciences, University of Johannesburg, Johannesburg 2006, South Africa; refilwep@uj.ac.za; 2Department of Statistics, Faculty of Science, University of Johannesburg, Johannesburg 2193, South Africa; yegnanews@uj.ac.za; 3TB HIV Care, Cape Town 8001, South Africa; hlengiwe@tbhivcare.org

**Keywords:** Consolidated Framework for Implementation Research, COVID-19, adolescent girls and young women, pre-exposure prophylaxis

## Abstract

**Background:** Innovative interventions, such as social media platforms and telemedicine, were implemented during the COVID-19 lockdown period for HIV prevention and treatment services. However, limited studies have reported on the facilitators and barriers of these innovations for HIV pre-exposure prophylaxis (PrEP) service continuity. Therefore, this study aimed to identify the barriers and facilitators of the implemented PrEP innovative interventions during COVID-19 among adolescent girls and young women (AGYW). **Methods:** A qualitative exploratory design was used to conduct semi-structured interviews with twelve stakeholders in the Dr Kenneth Kaunda District, North West Province of South Africa. Participants included various TB HIV Care programme stakeholders, comprising professional nurses, case managers, peer educators, and counsellors. The Consolidated Framework for Implementation Research (CFIR) 2.0 domains and constructs guided the interview questions and the analysis process. Additionally, all interviews were audio-taped, transcribed verbatim, and analyzed through thematic analysis. The facilitators and barriers of the PrEP innovative interventions were categorized according to the five CFIR domains. **Results:** The findings showed that despite the COVID-19 disruptions in healthcare services, the implemented innovative PrEP interventions enhanced the HIV prevention services. Facilitators included sufficient mobile data, teamwork, clear communication from managers, resilience, and existing media pages that supported social media-based PrEP service continuity. The implementation barriers included service users’ lack of cell phone devices, incorrect personal information, fear of contracting COVID-19, and limited individual movements. **Conclusion:** Social media and digital technologies played a crucial role in the continuation of HIV PrEP services among AGYW. These evaluations also illustrated the potential of social media platforms to be leveraged for HIV service delivery during periods of disruption, such as the COVID-19 lockdown period, for HIV service delivery. Furthermore, lessons learned from this study are significant and offer practical considerations for sustaining PrEP during service disruptions.

## 1. Introduction

According to the South African leading mathematical model of HIV, Thembisa Model, approximately 91,000 new HIV infections occurred in 2023 among women aged 15 years and older [1]. Oral pre-exposure prophylaxis (PrEP) is one of the most effective HIV prevention methods that has been explicitly implemented for “*adolescent girls and young women (AGYW)*” [2]. Daily oral PrEP involves taking antiretroviral medication to protect HIV-negative individuals at risk through sex or injection drug use [3]. For PrEP to be effective in maintaining a negative HIV serostatus, it requires strict adherence [4]. However, irrespective of its availability among AGYW, its uptake and adherence remain low among these populations, particularly during periods of disruption [5]. As such, side effects, stigma, and poor access contribute to PrEP discontinuation, underscoring the need for interventions that support consistent use among AGYW [2,6,7].

Studies from sub-Saharan Africa have shown that WhatsApp, SMS, and tele-PrEP can support HIV self-testing, adherence, and appointment scheduling during service disruptions [8,9]. Comparable evidence from outside the region indicates that telemedicine infrastructures sustained HIV prevention and PrEP care during the COVID-19 pandemic [10], while a global systematic review confirmed the feasibility and acceptability of virtual PrEP service delivery through mobile apps, telehealth, and SMS platforms [11]. Despite these innovations, systematic implementation science evaluations remain limited in African contexts. This study addresses that gap by applying CFIR 2.0 to assess the implementation of WhatsApp, Facebook, and tele-PrEP interventions for PrEP continuity during COVID-19 in the North West Province of South Africa.

The emergence of COVID-19 and its mitigation measures may have interrupted the consistent use of PrEP among AGYW, specifically during the alert level five measures, which limited individual movement and led to the closure of non-essential activities [12]. For instance, one study from Eswatini assessed PrEP uptake and adherence during and before the COVID-19 pandemic and found lower retention rates among service users aged 15–24 years [13]. Another study by Rao et al. [14] looked at the assessment of COVID-19’s impact on HIV prevention and care services among men who engage in sex with men (MSM) across twenty countries. Out of 9173 included MSM, 5171 reported interruptions to PrEP services during the lockdown period [14].

Responding to these service interruptions, novel approaches, such as social media platforms and telemedicine, were established as an urgent appointment system between doctors and patients [15]. Social media is “*a type of digital communication and interaction media produced by Web 2.0 technology development that facilitates interactions among individuals through which they create, share, or exchange information and ideas online in virtual communities and networks*” [16]. Furthermore, it can be acknowledged that efforts were made for HIV service continuity through the implementation of innovative interventions. However, frequently implemented health interventions often fail to produce meaningful outcomes for patient care [17].

Several implementation science frameworks have been applied to expand interventions that were found to be effective in enhancing public health impact. These are not limited to the “*Reach, Effectiveness, Adoption, Implementation, and Maintenance (RE-AIM) framework*” [18], *the “Consolidated Framework for Implementation Research” (CFIR)* [17], *and the “Exploration, Preparation, Implementation, and Sustainment (EPIS) framework”* [19]. The RE-AIM framework, for example, was applied in one study aimed at assessing the practicality of incorporating HIV prevention services, including PrEP, into a family planning setting over six months [20]. Effective improvements were highlighted in the study findings, whereby the rates of PrEP initiators and uptake increased from 10% in the first month to a peak of 65% in three months [20]. Another example was the application of the CFIR whereby the framework was applied for a rapid evaluation of the implementation strategy to expand mobile health (mHealth) use in HIV care settings [21]. The use of the framework enabled the researchers to identify facilitators and barriers to the implementation strategy, which were useful in expanding mHealth usage [21].

This study aimed to use the updated CFIR 2.0 to evaluate the barriers and facilitators of the “*implemented PrEP innovative interventions during the COVID-19 lockdown period among AGYW in Dr Kenneth Kaunda District in the North-West Province of South Africa*”.

## 2. Materials and Methods

### 2.1. Conceptual Framework

The CFIR is regarded as one of the most widely used frameworks in guiding the evaluation of various implemented health interventions [17]. It consists of thirty-nine constructs and five major domains, including (1) *intervention characteristics,* (2) *inner setting,* (3) *outer setting,* (4) *characteristics of the individual involved in the implementation, and* (5) *implementation process* [17]. To thoroughly identify the facilitators and barriers of the implemented PrEP innovative interventions, we paired the above-mentioned domains with implementation outcome constructs, namely “*acceptability, adoption, appropriateness, costs, feasibility, fidelity, and sustainability”* [22]. Moreover, the paired constructs enabled the researchers to understand various aspects that may have influenced the implementation outcomes of the innovative interventions. Using the CFIR framework, the researchers were able to identify the barriers and facilitators of the implemented social media platforms according to the stakeholders’ perspectives. Additionally, the application of this framework has increased the efficiency of research, the generalizability, and interpretability of research findings [23].

Figure 1 below illustrates how the five domains were paired with Proctor’s implementation outcome constructs.

### 2.2. Study Design and Setting

Using a qualitative exploratory research design, we explored the implementation experiences of the TB HIV Care stakeholders on the implemented innovative PrEP interventions during the COVID-19 lockdown period. Qualitative designs are research methods employed when the researcher has a limited understanding of the subject [24]. Therefore, they were suitable for this study because little was known about the facilitators and barriers of the implemented innovative PrEP interventions. This current study was conducted in the Dr Kenneth Kaunda district of the North West Province, South Africa. The district was selected because it contains several non-governmental organizations, such as the TB HIV Care and SHOUT-IT-NOW programmes, which provide HIV prevention and treatment services among key populations, including AGYW.

### 2.3. TB HIV Care Programme

The TB HIV Care primarily provides comprehensive healthcare services, including HIV prevention and treatment, to general and key populations, youth, and communities [25]. For the current study, the TB HIV Care programme was suitable because it had been operating even before the lockdown period, and it responded to the COVID-19 pandemic by implementing innovations such as social media platforms for HIV service continuity during the lockdown period [26]. Moreover, this programme is led by the South African government to offer PrEP to HIV-negative sex workers and young women, along with life-saving antiretroviral treatment (ART), adherence support, and a full range of sexual and reproductive health services [27]. Furthermore, the TB HIV Care programme was working as the implementing partner of the North West Department of Health to combat HIV and TB by 2030.

### 2.4. Sampling and Size

A purposive sample of twelve stakeholders from the TB HIV Care programme was included in this study. These stakeholders () comprised professional nurses, a case manager, peer educators, a peer coordinator, drivers, a social worker, a data capturer, and a general assistant. The sample provided a rich description of the events that occurred during the lockdown period because all participants were highly engaged in the implementation of innovative PrEP interventions.

### 2.5. Data Collection

The first author (OL) conducted semi-structured interviews in the TB HIV Care facility from 31 July to 30 August 2024. These interviews explored information on the innovative PrEP interventions implemented during the pandemic focusing on their facilitators, and barriers to their use. Additionally, the questions were guided by the paired CFIR domains, implementation outcome constructs, and the research objective. Stakeholders were invited for a one-on-one interview session, which lasted for about 30–45 min. Moreover, the primary language used for the interviews was English. Before interviews, stakeholders signed informed consent and gave permission to record the conversation. Table 1 shows how the semi-structured interview questions were asked using the paired CFIR domains and implementation constructs. The interviews continued until reaching the data saturation point, defined as the point where no new information emerged during the interview sessions [28]. In this regard, data saturation was reached at the tenth stakeholder. However, the researcher continued to interview an additional two stakeholders to confirm the saturation point.

### 2.6. Trustworthiness

To ensure the trustworthiness of this study, we applied Lincoln and Guba’s four criteria of credibility, transferability, dependability, and confirmability. Credibility was strengthened through independent coding by two researchers, followed by consensus discussions. In cases where agreement could not be reached, a third senior researcher adjudicated the discrepancies. An audit trail documenting coding decisions and theme development was maintained, and reflexive memos were kept acknowledging and managing researcher subjectivity, thereby enhancing confirmability. Transferability was supported through the provision of a thick description of the study context, participants, and setting. Dependability was ensured by peer debriefing sessions with the wider author team, where the identified barriers and facilitators were critically reviewed and refined to reach final agreement on the emergent themes.

### 2.7. Data Analysis

Data collection materials for analysis included the researcher’s reflection notes, field notes, and transcribed interview transcripts. A coding template was developed to identify the barriers and facilitators of the implemented interventions according to five CFIR domains, which may have affected the implementation phases. The interview transcripts were independently coded by two researchers with a range of qualifications, including a doctoral student and a research specialist. One of the coders was not involved in the data collection, which reduced potential subjectivity in interpreting the stakeholders’ experiences. The analysis was conducted manually by the two researchers and followed Braun and Clarke’s six steps of thematic analysis [29]:

(1) The researchers commenced by immersing themselves with the data by reading and re-reading the data, as well as noting initial ideas; (2) initial codes were generated and relevant data was organized according to each code; (3) then the codes were grouped into themes; (4) the coders met to review the emerged themes and to review them to ensure if they relate to the coded extract and the entire datasets; (5) each theme’s specifics were refined, which led to an ongoing analysis process, and the overall story of the analysis was told, generating clear definitions and names for each theme; and (6) a final report of the analysis was produced.

Furthermore, after the two coders reached a consensus on the identified barriers and facilitators of the implemented innovative PrEP interventions, further discussions occurred with all the authors to attain a final agreement on the emerged themes. In instances where the two coders were unable to resolve a discrepancy, a third senior researcher was consulted to adjudicate and facilitate consensus.

### 2.8. Ethical Considerations

Study ethics approval was secured from the University of Johannesburg’s Research Ethics Committee (UJREC) (REC-2435-2023) on 9 November 2023. Additional approvals were also obtained from the North West Department of Health (NW 202,311 010) and TB HIV Care before data collection [30]. During the recruitment stage, stakeholders were fully informed about the study and the expectations placed upon them. They received a research information letter that provided further details before signing the written informed consent forms. Additionally, stakeholders were assured that their participation was strictly voluntary during the data collection process. Furthermore, stakeholders’ privacy and confidentiality were upheld, as no identifiable references were included in their interview transcripts.

## 3. Results

A total of 12 semi-structured interviews were conducted with TB/HIV Care stakeholders, who represented various professional levels, including a case manager, two drivers (one of whom also served as a counsellor), a general assistant, two nurses, a social worker, a data capturer, three peer educators, and a peer coordinator. Table 2 the participants’ characteristics.

The identified themes were categorized under the paired CFIR domains, and the implementation outcome constructs, summarized in Table 3 and Table 4 below. These themes illuminate the barriers and enabling factors faced during the implementation of social media platforms, including WhatsApp Messenger and Facebook media pages for PrEP service continuity. Table 3 shows enabling factors; namely: sufficient mobile data, existing media pages, teamwork, clear communication, resilience, and mobile clinic vehicles. Collectively, these facilitators illustrate how resources and collaboration played a central role in sustaining service continuity during the COVID-19 lockdown.

Table 4 outlines the barriers that emerged during implementation. Among these were limited access to mobile devices, fear of contracting COVID-19, low PrEP uptake, provision of incorrect personal details, restrictions on individual movement, and lack of regular communication. Taken together, these barriers underscore the influence of the digital divide, the restrictive conditions of the pandemic, and the challenges of maintaining consistent engagement with service users.


**Domain 1: Intervention characteristics (Adoption, Acceptability, Feasibility, Cost)**


Several themes have emerged under the first domain, which indicate significant barriers and facilitators of the implemented social media platforms for the provision of PrEP services during the COVID-19 lockdown period.

1.1.Sufficient mobile data bundles

The importance of having enough mobile data was realized during the lockdown period, as social media platforms were utilized as a main form of individual interaction. Stakeholders acknowledged the role of the TB HIV Care Organisation in providing sufficient mobile data bundles to support the delivery of HIV PrEP services to service users.


*“Our organisation provided us with enough data bundles every month, so we did not struggle with mobile data”*
(Stakeholder 1, female, 27 years old)

1.2.Existing media pages

Services provided by the TB HIV Care stakeholders are normally offered through a mobile clinic vehicle within different communities; however, they also have a WhatsApp messenger that is always used for appointment schedules. During the lockdown period, the app, together with the Facebook page, was valuable because they were used as additional communication tools between service providers and users.


*“You see, sometimes it just depends on a person if she wants to live a healthy life, because we give them our contact details so that they can call/WhatsApp us anytime they need us”*
(Stakeholder 5, female, 50 years old)

1.3.Lack of mobile devices

Although there were facilitators under the first domain, the lack of mobile devices was indicated as a significant barrier to PrEP service continuity during the pandemic. Stakeholders mentioned that most of their service users are involved in transactional sex and drug use. Therefore, owning a mobile device is not their priority because, in most cases, they tend to sell them in exchange for drugs.


*“Not a lot of them remember our main challenge with our service users is that it is not common for they too have or own a mobile device for a long time, even though our tracing process is affected by this trend. Because some of them are drug users, if they need money, the first thing that will come to mind would be to sell their mobile devices”*
(Stakeholder 6, female, 37 years old)


**Domain 2: Outer setting (Penetration, Sustainability, and Adoption)**


The second domain focused more on contextual aspects that may positively or negatively influence the implementation of a new intervention. Under this theme, two barriers and one facilitator emerged.

2.1.Mobile clinic vehicle

Stakeholders indicated that, even though the country was in a lockdown state, they did manage to move around the communities and provide services with their mobile clinic vehicle. Through these movements, service users were able to recognize them and utilize the HIV provided services, and stakeholders were also giving out their social media details for further assistance.


*“My understanding is that most of them know us through our mobile clinic, cos now they can recognize us. Yeah, we did have our social media page for some time, so that assisted us throughout”*
(Stakeholder 2, male, 34 years old)

2.2.Fear of contracting COVID-19

One of the barriers under the outer setting theme was the fear of contracting the virus. According to the stakeholders, some of the service users were hesitant to meet with them for services within communities because of the fear of being infected with the Coronavirus disease 2019.


*“Eish, it was a struggle during those times because our clients did not want to meet with us because they were afraid of contracting COVID-19, and maybe the way we were dressed because we had to protect ourselves”*
(Stakeholder 4, female, 52 years old)

2.3.Low pre-exposure prophylaxis uptake

The second barrier was the low PrEP uptake and adherence, which resulted from the service users’ hesitation to involve themselves with stakeholders. Stakeholders highlighted that, irrespective of the implementation of innovative interventions, the COVID-19 restrictions impacted the PrEP services severely.


*“In terms of PrEP services, they were also slow because, in places like taverns, where we would find them mostly, it was not easy due to restrictions like social distancing”*
(Stakeholder 7, male, 30 years old)


**Domain 3: Inner setting (Feasibility, Fidelity, Sustainability)**


The implemented social media platforms were evaluated based on the Organizational capabilities. Under the third domain, two themes emerged on the facilitators and barriers.

3.1.Teamwork

One of the facilitators that kept the stakeholders going during the challenging COVID-19 times was their teamwork. Teamwork enabled stakeholders to implement and carry out the social media platforms for PrEP service continuity.


*“But during our team’s meetings, some suggestions have been raised in the implementation of some methods that will improve adherence and retention to care of our service users”*
(Stakeholder 9, female, 31 years old)

3.2.Incorrect personal details

False personal information containing residential addresses and cell phone numbers of service users was displayed as an important barrier to low service uptake. Stakeholders noted that, although the social media platforms seemed to be feasible and sustainable in the long run, much could not be achieved without reaching service users for regular follow-ups.


*“Uhm… untraceable, for instance. You will find that the contact details are not working, relocation, because our service providers depend on business; if there are no clients, they move to other places, and some don’t provide us with information about their whereabouts”*
(Stakeholder 6, female, 37 years old)


**Domain 4: Characteristics of individuals (Acceptability, Adoption, Fidelity)**


As we continued with the evaluation, stakeholders’ characteristics on the implemented innovative interventions were also imperative to be assessed, as they also play a crucial part in the success or failure of the new strategy. Underneath this domain, one facilitator and barrier emerged.

4.1.Clear communication channels

Stakeholders expressed that clear communication channels, such as regular team meetings on the COVID-19 updates, enabled them to adapt and accept the lockdown situation. The adaptability and acceptability allowed them the capabilities to implement and execute the innovative social media platforms for service continuity.


*“Every Monday, we have a team meeting where we discuss our schedules for the week. For example, we have two mobile clinic Vans that we use for services, which are maintained by two groups, so we will discuss which group is going where and also the challenges that we had last week, so that we can assist one another”*
(Stakeholder 8, female, 33 years old)

4.2.Limited individual movements

Due to COVID-19 mitigation measures like limited individual movements, stakeholders were unable to reach their service users, particularly when they were within different communities with the mobile clinic vehicle.


*“Yeah, I could say it did impact, I don’t want to say a lot, but it did make certain things a bit difficult because we couldn’t reach some of our service users, it was hard to get them given the fact that through certain stages certain people can’t come out things were so limited”*
(Stakeholder 1, female, 27 years old)


**Domain 5: Implementation process (Fidelity and Sustainability)**


The last domain is concerned with whether the intervention was implemented as planned and managed to reach the intended goals, as well as whether it can be sustained in the long term. The themes that emanated were one facilitator and one barrier.

5.1.Stakeholders’ resilience

One of the significant qualities of stakeholders during the implementation of social media platforms was their resilience to continue serving the service users regardless of the COVID-19 situation. Some of the stakeholders were not regular users of social media. However, they managed to learn and adopt those media pages for HIV service continuity.


*“Yoh, what can I say, we really tried to maintain services and learn to adapt as much as we could because we were worried that our organisation would close. So, we kept on pushing ourselves to recruit new clients and retain the old ones into care”*
(Stakeholder 10, female, 37 years old)

5.2.Lack of communication

Lack of regular communication from service users was identified as a barrier, which resulted in low PrEP uptake and adherence. Although social media apps such as WhatsApp were available for service users to interact with stakeholders regarding their PrEP refills, some users were not cooperating, which led to lost follow-up cases.


*“But some are committed to their treatment because they would call us to inform us that they moved, and then we can refer/link them to our nearby TB HIV Care, to where they are, because we also have branches in other areas too”*
(Stakeholder 11, female, 47 years old)

## 4. Discussion

The current study has demonstrated how the CFIR domains can be utilized to evaluate the implemented innovative PrEP interventions based on their barriers and facilitators. Facebook, WhatsApp, and telemedicine were crucial in conveying health information during the COVID-19 lockdown. Even after the pandemic, social media and digital technologies are evolving and remain integrated into nearly every aspect of daily life [31]. Due to their popularity and accessibility among adolescents and young adults, social media platforms have the potential to be leveraged to enhance PrEP services among AGYW. The current study provided an extensive description of the facilitators and barriers of the implemented social media platforms that were used to administer PrEP services among AGYW during the COVID-19 lockdown period using the CFIR framework.

The innovative social media platforms discussed in this study displayed various characteristics that contributed to their implementations, which were explained across the five CFIR domains. In the first domain (intervention characteristics), we found that factors like sufficient mobile data bundles provided to the stakeholders and their existing media pages facilitated the implementation of social media platforms for PrEP services. Another study conducted in Thailand showed the feasibility of telehealth follow-ups on the delivery of ART during the COVID-19 pandemic [10]. On the other hand, a study by McKay et al. [31] revealed that service users encountered challenges traversing technological systems in place to utilize virtual health platforms. These findings correspond with the barrier identified in the first domain of the current study, i.e., wherein service users lacked mobile devices. The absence of cell phones impeded the implementation of innovative approaches and resulted in low PrEP uptake and adherence. These findings suggest digital innovations should be evaluated for feasibility and usability before implementation, considering literacy and device access [8].

Looking at the second domain (outer setting), which focuses on how external factors influenced the implementation of social media platforms, the stakeholders’ existing mobile clinic promoted the intervention. However, service users’ fear of contracting the Coronavirus disease 2019 prevented the uptake of services. Similar findings suggest that patients avoided healthcare providers due to fear of infection exposure when seeking care at healthcare facilities [32]. Additionally, service providers in Kenya also appreciated that HIV testing services were made available during COVID-19 through novel approaches, but users remained at their homes due to fear of exposure, which resulted in the reduced uptake of HIV services [33]. These findings show how the COVID-19 mitigation measures disrupted both HIV prevention and treatment services.

The third domain focuses on inner settings, and it showed that stakeholders’ teamwork has positively impacted the employment of social media platforms. Teamwork has been labelled as the key factor in the success of any implementation. Because all the implementation efforts are dependent on teams, how they react to a new practice is likely to be influenced by team characteristics and processes [34]. Moreover, the importance of teamwork was also depicted in a study by Pinto et al. [35]. Service providers indicated that, although they faced personal setbacks during COVID-19, they were able to find creative solutions that led to the facilitation of HIV services availability under difficult circumstances [36].

Another important facilitator established in the fourth domain (characteristics of individuals) was the clear communication channels by the stakeholders’ managers that improved the implementation procedures. These findings corroborate the 2019–2022 evaluation report of the combination of HIV prevention interventions for AGYW by the HERStory Two study [9]. This study found that effective communication channels and trust between managers and frontline staff appeared to be key ingredients for successful implementation [9]. Despite the positive, clear communication channels, the implementation was hampered by the restricted individual movements during the lockdown period, which led to low PrEP outcomes. Contrary to the current study, Baron et al. [36] identified communication gaps as one of the barriers to PrEP delivery in South African rural areas in a CFIR-guided study, further highlighting the importance of leveraging digital platforms to improve communication, as successfully achieved in the current study. Bonnett et al. [37] reported the implementation of telehealth PrEP, which revealed both advantages, i.e., convenience, as well as disadvantages such as digital inaccessibility by some PrEP users. Further, Fischer et al. [38] reported the importance of context-specific mobile health interventions for improved PrEP uptake and adherence.

Furthermore, in the fifth domain (implementation process), stakeholders’ resilience was found to be a prominent factor in the implementation processes. Similarly, another CFIR-guided study reported stakeholder involvement and the continued assessment of the quality and progress of PrEP services as facilitators [39]. This is integral because, as mentioned by Wiig and colleagues, stakeholders’ resilience in healthcare is “the capacity to adapt to challenges and changes at different system levels, and to maintain high-quality care” [40]. A high level of resiliency was also observed among the community healthcare workers from Benin, Colombia, Guatemala, and Spain [41]. This emanated from their continued HIV service provision during COVID-19 among vulnerable populations [37]. These findings highlight the pivotal role of healthcare workers in maintaining the HIV care and prevention services during the pandemic.

Through these evaluations, the current study contributes to the lessons learned from the COVID-19 lockdown period to inform future pandemic preparedness. Moreover, studies reporting on the intervention evaluations are essential to understand the overall practicality of other PrEP interventions and the effectiveness of virtual service delivery models in low- and middle-income countries, including South Africa [11]. For instance, a study by Rousseau et al. [42] applied the market segmentation framework to improve PrEP interventions through targeted solutions and further reported the importance of this approach in improving the sustainability of PrEP use. Moreover, de Vos et al. [43] explored a behavioural framework inclusive of support for daily oral PrEP use and adherence among AGYW. However, there were still difficulties with adherence in this population despite the intervention.

## 5. Strengths and Limitations

The current study sheds light on the facilitators and barriers of the implemented social media platforms for HIV service continuity during the COVID-19 lockdown period. Its application to the CFIR domains offered a systematic process for the data collection and analysis of the study findings. Additionally, the CFIR domains enhanced the trustworthiness of the study. The study included a small sample size, which limited the findings from being generalized to the entire setting. Furthermore, the study was limited to only service providers since they were involved in the implementation of the novel approaches. The sample was limited to programme staff, excluding service users. This restricted the findings to implementers and highlights the need for future studies to include AGYW voices for a fuller understanding of PrEP delivery. Finally, the study is exploratory in nature and cannot be generalized to other settings. More studies are needed in various stakeholder groups and settings to yield more results.

## 6. Conclusions

Social media and digital technologies played a significant role in the continuation of HIV PrEP services among AGYW. Also, the assessments of these innovations have shown the potential of social media platforms to be leveraged to continue with HIV services during the COVID-19 lockdown period. Additionally, the identification of the barriers and facilitators sheds light on how these platforms can be improved for better PrEP outcomes for AGYW and can be applied in preparing for future pandemics. Although the study provided vital insights into the barriers and facilitators of the implemented PrEP innovative interventions during COVID-19 among AGYW, more related studies are needed to confirm and expand these findings.

## Figures and Tables

**Figure 1 ijerph-22-01602-f001:**
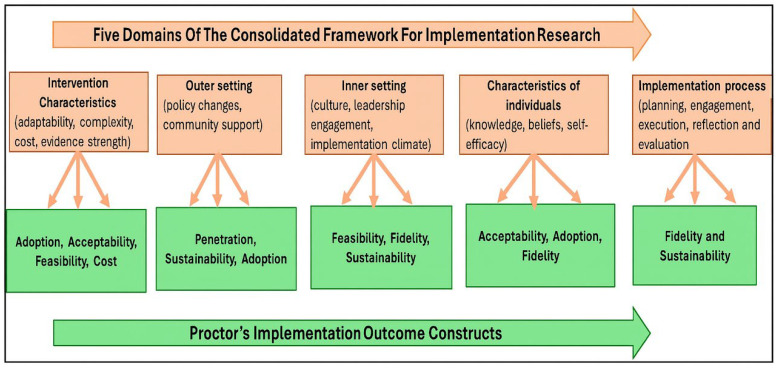
The Consolidated Framework Implementation Research’s five domains and implementation outcomes constructs. Adapted from [17,22].

**Table 1 ijerph-22-01602-t001:** Semi-structured interview questions.

CFIR Domain	Implementation Constructs	Semi-Structured Interview Questions
Intervention characteristics	Adoption, Acceptability, Feasibility, Cost	How did you perceive the implemented social media platforms for PrEP service continuity during the lockdown period? Were service users able to access them? Were there any extra resources to be added?
Outer setting	Penetration, Sustainability, Adoption	Were the interventions successful in reaching the target population? If not, what was the reason for this?
Inner setting	Feasibility, Fidelity, Sustainability	Were you able to implement these innovative PrEP interventions as planned? If not, why?
Characteristics of individuals	Acceptability, Adoption, Fidelity	Were you equipped efficiently to adapt the implemented innovative interventions into PrEP services?
Implementation process	Fidelity, Sustainability	If these innovative interventions were implemented as planned, are there any indicators suggesting that the intervention has the potential to be sustained?

**Table 2 ijerph-22-01602-t002:** Stakeholders’ characteristics.

Stakeholders	Age (Years)	Gender	Roles	Length of Service (Years)	Highest Qualification
**Stakeholder 1**	27	Female	Case manager	7	Secondary
**Stakeholder 2**	30	Male	Driver	2	Tertiary
**Stakeholder 3**	31	Female	General assistant	1	Secondary
**Stakeholder 4**	31	Female	Nurse	2	Tertiary
**Stakeholder 5**	33	Female	Nurse	2	Tertiary
**Stakeholder 6**	34	Male	Driver/counsellor	7	Tertiary
**Stakeholder 7**	37	Female	Social worker	6	Tertiary
**Stakeholder 8**	37	Female	Data capturer	7	Tertiary
**Stakeholder 9**	44	Female	Peer educator	4	Secondary
**Stakeholder 10**	47	Female	Peer coordinator	6	Secondary
**Stakeholder 11**	50	Female	Peer educator	4	Secondary
**Stakeholder 12**	52	Female	Peer educator	2	Secondary

**Table 3 ijerph-22-01602-t003:** Facilitators of the implemented PrEP innovative interventions.

CFIR Five Domains	Implementation Outcomes	Implemented Innovations	Facilitators
Intervention characteristics	Adoption, Acceptability, Feasibility, Cost	WhatsApp	Sufficient mobile data
		Facebook pages	Existing media pages
		Telemedicine	
Outer setting	Penetration, Sustainability, and Adoption	WhatsApp	Mobile clinic vehicle
		Facebook pages	
		Telemedicine	
Inner setting	Feasibility, Fidelity, Sustainability	WhatsApp	Teamwork
		Facebook pages	
		Telemedicine	
Characteristics of individuals	Acceptability, Adoption, Fidelity	WhatsApp	Clear communication channels
		Facebook pages	
		Telemedicine	
Implementation process	Fidelity, Sustainability	WhatsApp	Stakeholders’ resilience
		Facebook pages	
		Telemedicine	

**Table 4 ijerph-22-01602-t004:** Barriers to the implemented innovative PrEP interventions.

CFIR Domain	Implementation Outcomes	Implemented Innovations	Key Barriers
Intervention characteristics	Adoption, Acceptability, Feasibility, Cost	WhatsApp Facebook pages Telemedicine	Lack of mobile devices
Outer setting	Penetration, Sustainability, Adoption	WhatsApp Facebook pages Telemedicine	Fear of contracting COVID-19 Low PrEP uptake
Inner setting	Feasibility, Fidelity, Sustainability	WhatsApp Facebook pages Telemedicine	Incorrect personal information
Characteristics of individuals	Acceptability, Adoption, Fidelity	WhatsApp Facebook pages Telemedicine	Limited individual movement
Implementation process	Fidelity, Sustainability	WhatsApp Facebook pages Telemedicine	Lack of regular communication

## Data Availability

The raw data that supported the findings of this study are available from the corresponding author, L.L.O., upon fair and reasonable request.

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
