# Peer review of "A Qualitative Consolidated Framework for Implementation Research Evaluation of Innovative PrEP Delivery During COVID-19 Among Adolescent Girls and Young Women in North West Province, South Africa"

_ijerph, 2025, doi:10.3390/ijerph22101602_

Round 1
Reviewer 1 Report (New Reviewer)
Comments and Suggestions for Authors
As attached

Author Response
We would like to extend our sincere gratitude to the reviewers for their invaluable time and effort in reviewing the manuscript. Your insightful feedback and constructive comments have contributed to improving the revised version of this manuscript. The rebuttal letter contains comments and point-by-point responses, as well as the page numbers where the changes are inside the manuscript. The changes made are tracked changed inside the document manuscript. Thank you.
|
REVIEWER ONE |
||
|
Comment |
Response |
Page # |
|
Line 30-5: The facilitators should come before the barriers. |
Thanks for the input. We have aligned the sentences |
1 |
|
Line 52-55: The two sentences “Challenges like side effects, stigma, and access contribute to the discontinuation of PrEP [6-7]. Highlighting the need to implement effective interventions that can support the consistent use of PrEP among AGYW [2].” should be merged as each is meaningless as a stand-alone sentence. |
Thanks. We have combined the sentences |
2 |
|
Line 160: Where is the “coding template”? This can be included as a supplementary file. |
Thank you for the comment. The coding template/framework has been attached as supplementary material. |
Supplementary file 1 |
|
Line 176: How were cases where the coders could not reach a consensus resolved? |
We have clarified that a third researcher was consulted to facilitate consensus in such cases |
6 |
|
Line 195: Table 2 can be re-arranged as follows: Stakeholders Age (Years) Gender Roles Length of service (Years) Highest qualification |
Thanks for this comment. We have fixed the headings |
7 |
|
Under the Length of service, the authors should just state the number without including years, e.g., “7” not “7 years.” |
Thank you for the comment. We have deleted the text |
7 |
|
Line 351-2: This sentence is not clear. |
The sentence has been clarified as follows: These findings correspond with the barrier identified in the first domain of the current study, i.e., wherein service users lacked mobile devices.’’ |
12 |
|
The authors made some good attempts to discuss the results. To consolidate your discussion, I would suggest the authors should compare frameworks in other climes used to evaluate pre-exposure prophylaxis interventions. |
Thank you for the comment. More comparisons were made with other studies that applied different frameworks as follows: Contrary to the current study, Baron et al. [37] identified communication gaps as one of the barriers to PrEP delivery in South African rural areas in a CFIR-guided study, further highlighting the importance of leveraging digital platforms to improve communication, as successfully done in the current study. Bonnett et al. [38] reported the implementation of telehealth PrEP, which revealed both advantages, i.e, convenience, as well as disadvantages such as digital inaccessibility by some PrEP users. Further, Fischer et al. [39] reported the importance of context-specific mobile health interventions for improved PrEP uptake and adherence. Furthermore, in the fifth domain (implementation process), stakeholders’ resilience was found to be a prominent factor in the implementation processes. Similarly, another CFIR-guided study reported stakeholder involvement and the continued assessment of the quality and progress of PrEP services as facilitators [40]. For instance, a study by Rousseau et al. [43] applied the market segmentation framework to improve PrEP interventions through targeted solutions, and further reported the importance of this approach in improving the sustainability of PrEP use. Moreover, de Vos et al [44] explored a behavioural framework inclusive of support for daily oral PrEP use and adherence among AGYW. |
13 |
|
Line 406: The strengths and weaknesses should come under the discussion section. |
Thanks for the input. We have aligned the sections |
13-14 |
Reviewer 2 Report (New Reviewer)
Comments and Suggestions for Authors
|
Manuscript section |
Key issues |
Specific, actionable suggestions |
|
Title & Abstract |
Title omits study design and setting.
 Abstract over‑reaches in its final sentence (“can be utilised for future pandemic preparedness”) without linking to data. |
Re‑title as “A qualitative CFIR evaluation of innovative PrEP delivery during COVID‑19 among AGYW in North‑West Province, South Africa”.
 Replace the last Abstract line with a statement that strictly reflects your data (e.g., “Findings offer practical considerations for sustaining PrEP during service disruptions.”). |
|
Introduction |
Background heavy on PrEP biology; sparse on digital‑health evidence in sub‑Saharan Africa.Reference list skews to global/Western studies. |
Insert 1‑2 paragraphs summarising regional evidence on WhatsApp / tele‑PrEP (e.g., Rogers 2020; Patel 2022) and clarify the unique knowledge gap your study addresses. |
|
Conceptual framework |
Figure 1 duplicates CFIR diagram but does not explain how Proctor constructs were operationalised in coding. |
Add a short paragraph (or supplementary table) mapping each Proctor construct to at least one interview question and one analytic code. |
|
Methods – sampling |
Purposive sample limited to programme staff; no service‑user voices. Insufficient justification for sample size beyond “saturation”. |
Acknowledge the single‑perspective limitation in Limitations.Provide a brief audit trail: total staff eligible = N, invited = N, declined = N. |
|
Methods – data collection |
Interview guide not supplied; language translation process unclear. |
Add the full guide and a summary of translation/back‑translation steps as Online Appendix A. |
|
Methods – analysis |
Manual coding; intercoder reliability not reported. Reflexivity statement absent. |
Report how coding disagreements were resolved and provide a simple κ or percentage agreement.
Add 2‑3 lines on researchers’ positionality and how it was managed. |
|
Results |
Themes well organised but few illustrative quotes.Table 3 mixes CFIR constructs and outcomes; hard to track evidence chain. |
Insert 1–2 verbatim quotes per theme to strengthen credibility.
Split Table 3 into two clearer tables: (1) barriers, (2) facilitators, each traced to a specific CFIR construct. |
|
Discussion |
Some claims (e.g., “social media platforms have potential beyond pandemics”) are not linked back to data. Minimal comparison with other CFIR‑based digital‑health evaluations in LMICs. |
Tighten causal language (“suggests” instead of “shows”). Add a paragraph contrasting your findings with at least two other CFIR‑guided digital interventions in SSA. |
|
Conclusions |
Over‑generalises from 12 staff. |
Temper phrasing: emphasise exploratory nature and need for multi‑stakeholder replication. |
|
Figures & tables |
Figure 1 low resolution; no legend for acronyms.
Table 2 headers mis‑aligned. |
Supply a high‑resolution, stand‑alone Figure 1 with acronym key.
Re‑format Table 2 for readability. |
|
Language & style |
Minor grammatical slips (“mitigation measures disrupted both HIV prevention and treatment services”). |
Recommend a light professional copy‑edit. |
Comments on the Quality of English Language
The English could be improved to more clearly express the research.
Author Response
We would like to extend our sincere gratitude to the reviewers for their invaluable time and effort in reviewing the manuscript. Your insightful feedback and constructive comments have contributed to improving the revised version of this manuscript. The rebuttal letter contains comments and point-by-point responses, as well as the page numbers where the changes are inside the manuscript. The changes made are tracked changed inside the document manuscript. Thank you.
|
Comment |
Response |
Page # |
|
REVIEWER 2 |
||
|
Re title as “A qualitative CFIR evaluation of innovative PrEP delivery during COVID 19 among AGYW in North West Province, South Africa”.
 |
The title of the study has been revised as suggested, it now reads as follows: ‘’A qualitative Consolidated Framework for Implementation Research evaluation of innovative PrEP delivery during COVID-19 among Adolescent Girls and Young Women in North West Province, South Africa’’.
 |
1 |
|
Replace the last Abstract line with a statement that strictly reflects your data (e.g., “Findings offer practical considerations for sustaining PrEP during service disruptions.”). |
The sentence has been replaced as advised. |
1 |
|
Insert 1 2 paragraphs summarising regional evidence on WhatsApp / tele PrEP (e.g., Rogers 2020; Patel 2022) and clarify the unique knowledge gap your study addresses. |
The paragraph has been added as advised. It now reads as follows: Studies from sub-Saharan Africa have shown that WhatsApp, SMS, and tele-PrEP can support HIV self-testing, adherence, and appointment scheduling during service disruptions [8,9]. Comparable evidence from outside the region indicates that telemedicine infrastructures sustained HIV prevention and PrEP care during the COVID-19 pandemic [10], while a global systematic review confirmed the feasibility and acceptability of virtual PrEP service delivery through mobile apps, telehealth, and SMS platforms [11]. Despite these innovations, systematic implementation science evaluations remain limited in African contexts. This study addresses that gap by applying CFIR 2.0 to assess the implementation of WhatsApp, Facebook, and tele-PrEP interventions for PrEP continuity during COVID-19 in the North-West Province of South Africa. |
2 |
|
Add a short paragraph (or supplementary table) mapping each Proctor construct to at least one interview question and one analytic code. |
Thank you for the comment. This has been added as supplementary material. |
Supplementary file 1 |
|
Acknowledge the single perspective limitation in Limitations. Provide a brief audit trail: total staff eligible = N, invited = N, declined = N. |
The limitation section has been revised. |
13-14 |
|
Add the full guide and a summary of translation/back translation steps as Online Appendix A. |
Thank you for the comment. Although the courtesy of using other local languages that the stakeholders were comfortable in was extended, all participants managed to express themselves in English, and all the included verbatim codes are in English. This was revised for clarity in the data collection section. Hence, there was no translation necessary. |
4 |
|
Report how coding disagreements were resolved and provide a simple κ or percentage agreement.
Add 2 3 lines on researchers’ positionality and how it was managed. |
We have clarified how disagreements were resolved by the third researcher/coder. |
6 |
|
Insert 1–2 verbatim quotes per theme to strengthen credibility.
 |
Thank you for the comment. For each of the themes, one quotation that was observed to represent the theme best was selected to avoid tautology. |
6-12 |
|
Split Table 3 into two clearer tables: (1) barriers, (2) facilitators, each traced to a specific CFIR construct. |
The comment has been addressed as suggested. Table 3 was split into two, i.e., Table 3 and 4 for facilitators and barriers, respectively. |
Pages 8-9 |
|
Tighten causal language (“suggests” instead of “shows”). Add a paragraph contrasting your findings with at least two other CFIR-guided digital interventions in SSA. |
The paper has undergone language editing, and all causal language was removed/improved. Comparisons were made with other studies. See below: ‘’Contrary to the current study, Baron et al.[36] identified communication gaps as one of the barriers to PrEP delivery in South African rural areas in a CFIR-guided study, further highlighting the importance of leveraging digital platforms to improve communication, as successfully done in the current study. Bonnett et al. [37] reported the implementation of telehealth PrEP, which revealed both advantages, i.e, convenience, as well as disadvantages such as digital inaccessibility by some PrEP users. Further, Fischer et al. [38] reported the importance of context-specific mobile health interventions for improved PrEP uptake and adherence.’’ |
Entire manuscript
13 |
|
Temper phrasing: emphasise exploratory nature and need for multi stakeholder replication. |
Thank you for the comment, in addition to the explaratory nature of the study being mentioned in the abstract and study design sections, this has further been clarified on page 11 in the limitations and conclusions of the study respectively, as follows: " Last but not least, the study is exploratory in nature and cannot be generalised to other settings. More studies are needed in various stakeholder groups and settings to yield more results. ‘’Although the study provides vital insights into the barriers and facilitators of the implemented PrEP innovative interventions during COVID-19 among AGYW, more related studies are needed to confirm and expand these findings. ‘’ |
13-14 |
|
Supply a high-resolution, stand alone Figure 1 with acronym key.
Re format Table 2 for readability. |
Both Table 2 and Figure 1 have been revised as suggested. An acronym key could not be included since there are no acronyms in the figure. |
4 and 7 |
|
Recommend a light professional copy edit. |
The manuscript has been language edited as suggested. |
Entire manuscript |
Reviewer 3 Report (New Reviewer)
Comments and Suggestions for Authors
This is worthy, timely and relevant research during COVID when social interactions were impacted and potentially life-threatening to the sample. The article aimed to use the updated Consolidated Framework for Implementation Research to evaluate pre-exposure prophylaxis interventions during COVID-19 in the North-West Province in South Africa. This is an appropriate qualitative exploratory research design given the nature of the topic.
I will comment on the guidelines provided:
- Is the manuscript clear, relevant for the field and presented in a well-structured manner? YES
- Are the cited references mostly recent publications (within the last 5 years) and relevant? YES. Does it include an excessive number of self-citations? NO
- Is the manuscript scientifically sound and is the experimental design appropriate to test the hypothesis? YES
- Are the manuscript’s results reproducible based on the details given in the methods section? YES
- Are the figures/tables/images/schemes appropriate? Do they properly show the data? YES Are they easy to interpret and understand? Is the data interpreted appropriately and consistently throughout the manuscript? YES
- Please include details regarding the statistical analysis or data acquired from specific databases. I WILL COMMENT ON THIS BELOW
- Are the conclusions consistent with the evidence and arguments presented? YES
- Please evaluate the ethics statements and data availability statements to ensure they are adequate. ADEQUATE
- Line 136 notes the range of stakeholders in plural form eg. social workers but there is only one in the sample.
- It would have been helpful for a qualitative study to have some indication of the researchers' positionality, their reflection notes or field notes.
- No mention is made of the dimensions pertinent to qualitative methodology eg. credibililty, transferability (not generalizabilty), trust worthiness, reflexivity etc.
- Line 190 the social worker is not listed among the professional levels although listed in the table in line 195 ( Stakeholder 7)
- Line 258 it would be interesting to know if stigma as a barrier played any role in service users not accessing services during COVD-19.
- Line 303. The aim of the study was to explore the feedback from service providers but this study would have been enhanced by exploring the feedback from service users. This may be considered for future research and will make a great contribution to service delivery. The researchers do point out that the exclusion of service users was a limitation.
- See line 336,337 about possible role of stigma given fear of exposure to infection.
Author Response
We would like to extend our sincere gratitude to the reviewers for their invaluable time and effort in reviewing the manuscript. Your insightful feedback and constructive comments have contributed to improving the revised version of this manuscript. The rebuttal letter contains comments and point-by-point responses, as well as the page numbers where the changes are inside the manuscript. The changes made are tracked changed inside the document manuscript. Thank you.
|
REVIEWER 3 |
||
|
Line 136 notes the range of stakeholders in plural form eg. social workers but there is only one in the sample. |
We have updated the section to reflect the correct number of staff categories |
4 |
|
It would have been helpful for a qualitative study to have some indication of the researchers' positionality, their reflection notes or field notes. |
Thank you for the comment. The researcher’s notes can be found on the coding template/framework document attached as a supplementary file. |
Supplementary file 1 |
|
No mention is made of the dimensions pertinent to qualitative methodology eg. credibililty, transferability (not generalizabilty), trust worthiness, reflexivity etc. |
We have added the Trustworthiness section. |
5-6 |
|
Line 190 the social worker is not listed among the professional levels although listed in the table in line 195 ( Stakeholder 7)
|
Thank you for the comment. The professional levels have been revised to include the social worker. |
6 |
|
Line 258 it would be interesting to know if stigma as a barrier played any role in service users not accessing services during COVD-19. |
Thank you for the comment. Stigma was not identified as a barrier to PrEP uptake in the current study during COVID-19 (see the identified barriers on Table 4), but rather the fear of AGYW contracting COVID-19. Additionally, the closure of hotspots where AGYW are usually found such as taverns were not accessible during COVID-19 as stated in 2.2 and 2.3 in the results section of the manuscript. |
9 |
|
Line 303. The aim of the study was to explore the feedback from service providers but this study would have been enhanced by exploring the feedback from service users. This may be considered for future research and will make a great contribution to service delivery. The researchers do point out that the exclusion of service users was a limitation. |
The study's limitation was acknowledged. |
13-14 |
|
See line 336,337 about possible role of stigma given fear of exposure to infection.
|
Thank you for the comment. Sigma was not identified by the current study as a barrier to accessing PrEP services during COVID-19 (see the barriers identified in Table 4). However, fear of exposure to COVID-19 was identified as a major barrier. |
7 and 9 |
Round 2
Reviewer 2 Report (New Reviewer)
Comments and Suggestions for Authors
Comments for Authors
The authors have done a commendable job addressing the round 1 feedback. Specific notes:
Title/Abstract
- Title revised appropriately to include study design and setting.
- Abstract conclusion now tempered and reflects findings rather than over-reaching. This aligns well with reviewer advice.
Introduction
- Added regional and global literature (e.g., Rogers 2020; Patel 2022; systematic reviews on tele-PrEP)
- The gap in systematic implementation science evaluations in African contexts is now clearly justified.
Methods
- Inclusion of supplementary coding framework, audit trail, reflexivity, and coder agreement strengthens transparency.
- Clarification of saturation (achieved at 10 interviews, confirmed with 12) and ethics approvals is clear.
- Trustworthiness section (credibility, dependability, confirmability, transferability) is a valuable addition.
Results
- Verbatim quotes per theme included, though usually one per theme; this is acceptable for brevity but could limit richness.
- Table 3 and 4 separation improves readability
- Findings align with CFIR domains, structured and coherent.
Discussion
- Overly causal language replaced with cautious terms ("suggests," "indicates").
- Comparative discussion with other CFIR-guided digital interventions in SSA included (Baron et al., Bonnett et al., Fischer et al.)
- Limitations now acknowledge single perspective (provider-only), small sample, and non-generalizability.
Figures/Tables
- Figure 1 reformatted; tables clearer.
- Acronym clarification minor but adequate.
Language
- Substantial improvement since V1. Still, some sentences remain long and could be shortened for concision. For example:
- “The implemented social media platforms were also evaluated based on the Organizational capabilities…” could be simplified.
Minor revisions recommended: final light copy-edit for conciseness and consistency (esp. tense alignment, sentence length).
Author Response
Please see the attachment.

This manuscript is a resubmission of an earlier submission. The following is a list of the peer review reports and author responses from that submission.
Round 1
Reviewer 1 Report
Comments and Suggestions for Authors
The paper presents 12 CFIR-guided interviews based assessment of social-media and telemedicine interventions in facilitating HIV PrEP services during COVID-19.
However major revisions are needed in the paper. My detailed comments are as shown below:-
Title:
‘Using the Consolidated Framework for Implementation Research to assess pre-exposure prophylaxis interventions during COVID-19 in the North-West Province, South Africa’.
…assess pre-exposure intervention… - this is confusing.
This work only ‘qualitatively’ assesses the implementation and perceived success of the social media & telemedicine interventions.
Abstract & Main Text:
- Line 17 (also Line 68-69): ‘HIV Service continuity’.
Are the authors referring to the HIV care continuum here? However, this works only refers to PrEP continuum.
- Lines 18-20: ‘Therefore, this study aimed to explore the implementation experiences of TB HIV Care stakeholders on the implemented pre-exposure prophylaxis (PrEP) innovative interventions during COVID-19 among adolescent girls and young women (AGYW)’.
This sentence is unclear. Please simplify and restructure.
- Line 29-32: ‘The facilitators were identified as having sufficient mobile data bundles, stakeholders’ teamwork, clear communication channels from the managers, team resilience, and existing media pages, which were significant facilitators in implementing social media platforms for PrEP service continuity’.
There is a syntax error here. Please correct the sentence.
- Lines 32-33: ‘In conclusion, social media and digital technologies played a crucial role in the continuation of HIV PrEP services among AGYW’.
And
Line 33-35: ‘Also, the assessments of these innovations have shown the potential of social media platforms to be leveraged to continue with HIV services during the COVID-19 lockdown period’.
These conclusions here are not corroborated through the abstract’s results section.
Additionally, significant portion of 2nd sentence is redundant.
‘the assessments of these innovations’ – this suggests quantitation. However, this is a qualitative study, and the usage of assessments is misleading.
- The abstract requires significant revision. Per the IJERPH –
The abstract should be a single paragraph and should follow the style of structured abstracts, but without headings: 1) Background: Place the question addressed in a broad context and highlight the purpose of the study; 2) Methods: Describe briefly the main methods or treatments applied. Include any relevant preregistration numbers, and species and strains of any animals used; 3) Results: Summarize the article's main findings; and 4) Conclusion: Indicate the main conclusions or interpretations. The abstract should be an objective representation of the article: it must not contain results which are not presented and substantiated in the main text and should not exaggerate the main conclusions.
- Line 41: ‘…about 91,000 new HIV infections were among women aged 15 years and older’.
Line 42: ‘…Oral pre-exposure prophylaxis (PrEP) is among the powerful HIV prevention methods…’
Please correct the syntax errors here.
- Line 57: ‘…lower retention rates among clients…’
Usage of clients, in healthcare context, is inappropriate. Please change of subjects or patients.
- Line 60: ‘…perceived interruptions to PrEP services…’
What does ‘perceived’ interruptions refer to? This is confusing.
- Line 70: ‘However, often-implemented health interventions fail to provide significant client care outcomes’.
This is unclear. What does refer to?
- Line 85: What is mHealth?
- Line 86-87: ‘…describe the implementation experiences of TB HIV Care stakeholders, facilitators and barriers of the implemented PrEP innovative interventions…’
What does this mean? Please correct the syntax error here.
- Section 2.2, 2.3:
- What is referred to as TB HIV Care (also Lines 18-20, line 87) – ‘Tuberculosis’ HIV Care?
Why were the Tuberculosis HIV Care stakeholders specifically considered, and not the HIV care continuum team where PrEP is more specifically implemented as a framework.
This choice demands to be specifically addressed, beyond the explanation in lines 122-124. Given the TB HIV care is a subset of the larger HIV care continuum, selecting stakeholders only from a subset of the team biases the outcomes and experiences. This needs to be addressed both in the methods and expanded in the discussion sections.
- The interviewee’s quotes provided across the sections 1.1 – 5.2 is presented in continuation of the results. Please present them instead as Stakeholder/Interviewee’s Quote.
- Line 301: Please correct ‘WhatsApp apps’.
- A key issue with this work is that the results section do not provide a concise and precise description of the experimental results their interpretation as well as the experimental conclusions that can be drawn.
A. The n number included in this study is significantly low and does not allow the ability to draw any critical discussion and conclusions (as the authors claim in the abstract, discussion or conclusion sections).
B. Authors should consider assigning rating to the CFIR construct to allow quantitation of some key outcomes, including consolidation and summarization of the facilitators and barriers.
Author Response
Dear Reviewer,
I hope you are well.
Thank you for taking the time to review the manuscript.
I have attached the rebuttal report and revised manuscript.
Best regards,

Reviewer 2 Report
Comments and Suggestions for Authors
The manuscript is very interesting and highlights social media and digital technologies' critical role in continuing HIV PrEP services among AGYW. It also identifies facilitators and barriers to access to HIV PrEP services by this group.
The paper is well-written, objective, and transparent. In this way, the suggestions clarify the text, especially when read by non-South Africans.
1. It is essential to characterize the institution - TB HIV Care - the population assisted and its relation with the North-West Department of Health;
2. In the methodology, the authors affirm " (...) the primary language used for the interviews was English; however, to enhance participation comfort, stakeholders were also encouraged to use Setswana, the other common language in that setting." Were the interviewers fluent in Setswana?
3. In Table 2, in the same sense as the suggestion above, describe the highest qualification categories to facilitate the comprehension of non-South African readers.
4. It is important to identify who said each line used in the pieces of evidence.
5. It should be highlighted that the research didn´t interview the users of TB HIV Care and that it is limited to the stakeholders' perceptions.
Author Response

(The authors gave the same response as above.)

Reviewer 3 Report
Comments and Suggestions for Authors
Dear author, after reviewing the article entitled: Using the Consolidated Framework for Implementation Re-2 search to assess pre-exposure prophylaxis interventions during COVID-19 in the North-West Province, South Africa, I am sending you my suggestions and comments: Regarding the methodological structure of the study, there is a deficiency in the methodology used, since the study objective was to explore the implementation experiences of TB and HIV care stakeholders on the implementation of innovative pre-exposure prophylaxis (PrEP) interventions during COVID-19 among adolescent girls and young women. According to the proposed methodology, only health personnel and personnel who provided care and prevention strategies (i.e., those who implemented the strategy) were included in the study. However, it cannot be concluded whether the intervention was adequately implemented since those who received the information were not part of this study. The parameters evaluated could not have been analyzed by health personnel, but rather by those who received the intervention.
The program was applied not only to a population at high risk for HIV but also to those with tuberculosis, which is biased and inconsistent with the topic and objectives of this study. The methodology does not explain how the data analysis was conducted, whether a program or software was used for the analysis, whether data nodes were analyzed, whether the information was cross-referenced among researchers, what criteria were used to determine that the topic was exhausted, etc.
The discussion is sparse and does not allow for validation of the results obtained, since it is largely limited to describing the results already described rather than cross-referencing and validating the results.
This is all I can share with you after my review.
Best regards.
Comments on the Quality of English LanguageThe quality of the English translation of the article should be reviewed, as it makes the translated text difficult to read and understand.
Author Response

(The authors gave the same response as above.)
